# Assessment of the *In Vitro* and *In Vivo* Antifungal Activity of NSC319726 against *Candida auris*

Jizhou Li,[a,b] Alix T. Coste,[a] Daniel Bachmann,[a] Dominique Sanglard,[a] Frederic Lamoth[a,b]

[a]Institute of Microbiology, Lausanne University Hospital and University of Lausanne, Lausanne, Switzerland
[b]Infectious Diseases Service, Department of Medicine, Lausanne University Hospital and University of Lausanne, Lausanne, Switzerland

**ABSTRACT** *Candida auris* is an emerging yeast pathogen of candidemia with the ability to develop resistance to all current antifungal drug classes. Novel antifungal therapies against *C. auris* are warranted. NSC319726 is a thiosemicarbazone with an inhibitory effect on fungal ribosome biogenesis that has demonstrated some antifungal activity. In this study, we assessed the *in vitro* activity and *in vivo* efficacy of NSC319726 against *C. auris*. NSC319726 was active *in vitro* against 22 *C. auris* isolates from different clades, with MICs ranging from 0.125 to 0.25 mg/liter. Despite complete visual growth inhibition, the effect was described as fungistatic in time-kill curves. Interactions with fluconazole, amphotericin B, and micafungin, as tested by the checkerboard dilution method, were described as indifferent. NSC319726 demonstrated significant effects in rescuing *G. mellonella* larvae infected with two distinct *C. auris* isolates, compared to the untreated group. In conclusion, NSC319726 demonstrated *in vitro* activity against *C. auris* and *in vivo* efficacy in an invertebrate model of infection. Its potential role as a novel antifungal therapy in humans should be further investigated.

**IMPORTANCE** *Candida auris* is emerging as a major public health threat because of its ability to cause nosocomial outbreaks of severe invasive candidiasis. Management of *C. auris* infection is difficult because of its frequent multidrug-resistant profile for currently licensed antifungals. Here, we show that the thiosemicarbazone NSC319726 was active *in vitro* against a large collection of *C. auris* isolates from different clades. Moreover, the drug was well tolerated and effective for the treatment of *C. auris* infection in an invertebrate model of *Galleria mellonella*. We conclude that NSC319726 might represent an interesting drug candidate for the treatment of *C. auris* infection.

**KEYWORDS** antifungal resistance, antifungal susceptibility testing, candidiasis, thiosemicarbazone

Candida auris is an emerging yeast pathogen causing nosocomial outbreaks of invasive candidiasis (1–4). Five genetic clades have been identified, with distinct geographical origins, i.e., South Asian (clade I), East Asian (clade II), South African (clade III), South American (clade IV), and Iranian (clade V) (2, 5, 6). Acquired antifungal resistance is a hallmark of *C. auris* and can affect all three current antifungal classes (azoles, polyenes, and echinocandins) (2). More than 90% of *C. auris* isolates are resistant to fluconazole, and resistance to two or more antifungal classes is not uncommon (2, 7, 8). Therefore, the development of new antifungal drugs against *C. auris* is warranted.

NSC319726 is a thiosemicarbazone zinc chelator that inhibits the growth of mammalian cancer cell lines with a p53 mutation (9). This compound also demonstrated antifungal activity against several *Candida* species, *Cryptococcus neoformans*, and *Aspergillus fumigatus* (10). The aim of this study was to measure the antifungal activity of NSC319726 against a collection of 22 clinical isolates of *C. auris*, in comparison to standard antifungals, and to assess its efficacy in an invertebrate model of *C. auris* infection.

Address correspondence to Frederic Lamoth, frederic.lamoth@chuv.ch.

**TABLE 1** MICs of NSC319726 and standard antifungal drugs against 22 *Candida auris* clinical isolates

| Strain | Clade/origin (reference) | MIC ($\mu$g/ml) | | | | |
| --- | --- | --- | --- | --- | --- | --- |
| | | Fluconazole | Voriconazole | Amphotericin B | Micafungin | NSC319726 |
| I.2 | I/India (16) | >64 | 1 | 1 | 0.125 | 0.125 |
| I.3 | I/India[a] | >64 | 2 | 1 | >8 | 0.25 |
| I.4 | I/India (20) | >64 | 1 | 1 | 0.125 | 0.25 |
| II.1 | II/Japan (15) | 4 | 0.0625 | 0.5 | 0.125 | 0.125 |
| II.2 | II/India[a] | 32 | 1 | 0.25 | 0.125 | 0.125 |
| III.7 | III/Switzerland (17) | >64 | 2 | 0.5 | 0.25 | 0.25 |
| III.8 | III/Israel (18) | >64 | 2 | 0.5 | 0.25 | 0.125 |
| III.9 | III/Israel (18) | >64 | 2 | 1 | 0.25 | 0.25 |
| IV.1 | IV/Colombia (19) | 4 | 0.0625 | 2 | 0.125 | 0.125 |
| IV.2 | IV/Colombia (19) | >64 | 1 | 2 | 0.125 | 0.25 |
| IV.3 | IV/Colombia (19) | >64 | 1 | 2 | 0.125 | 0.125 |
| IV.4 | IV/Colombia (19) | >64 | 0.5 | 1 | 0.125 | 0.125 |
| IV.5 | IV/Colombia (19) | >64 | 0.5 | 1 | 0.125 | 0.125 |
| IV.6 | IV/Colombia (19) | >64 | 0.5 | 1 | 0.125 | 0.125 |
| IV.7 | IV/Colombia (19) | >64 | 8 | 1 | 0.25 | 0.25 |
| IV.8 | IV/Colombia (19) | >64 | 8 | 1 | 0.25 | 0.25 |
| IV.9 | IV/Israel (18) | 16 | 1 | 0.5 | 0.25 | 0.25 |
| IV.10 | IV/Israel (18) | 32 | 0.5 | 0.25 | 0.25 | 0.25 |
| IV.11 | IV/Israel (18) | 32 | 0.5 | 0.5 | 0.25 | 0.25 |
| IV.12 | IV/Israel (18) | 32 | 0.5 | 0.5 | 0.25 | 0.25 |
| IV.13 | IV/Israel (18) | 32 | 0.5 | 1 | 0.25 | 0.125 |
| IV.14 | IV/Israel (18) | 32 | 1 | 0.5 | 0.5 | 0.25 |

[a]A gift from Maurizio Sanguinetti.

## RESULTS

MICs obtained by the broth microdilution method (11) are shown in Table 1. The $MIC_{50}$ and $MIC_{90}$ values (i.e., values encompassing 50% and 90% of isolates, respectively) for fluconazole, voriconazole, amphotericin B, and micafungin were >64 and >64 $\mu$g/ml (range, 4 to >64 $\mu$g/ml), 1 and 2 $\mu$g/ml (range, 0.06 to 8 $\mu$g/ml), 1 and 2 $\mu$g/ml (range, 0.25 to 2 $\mu$g/ml), and 0.25 and 0.25 $\mu$g/ml (range, 0.125 to >8 $\mu$g/ml), respectively. According to the tentative breakpoints of the Centers for Disease Control and Prevention (CDC) (https://www.cdc.gov/fungal/candida-auris/c-auris-antifungal.html), 19, 3, and 1 of these 22 isolates were defined as resistant to fluconazole (MIC breakpoint, ≥32 $\mu$g/ml), amphotericin B (MIC breakpoint, ≥2 $\mu$g/ml), and micafungin (MIC breakpoint, ≥4 $\mu$g/ml), respectively.

NSC319726 was active against all 22 *C. auris* strains, with $MIC_{50}$ and $MIC_{90}$ values of 0.25 and 0.25 $\mu$g/ml, respectively (range, 0.125 to 0.25 $\mu$g/ml) (Table 1). In order to better characterize the antifungal effect of NSC319726, further experiments were performed using two selected isolates from two different clades, i.e., one azole- and echinocandin-susceptible isolate (IV.1) and one azole- and echinocandin-resistant isolate (I.3). Time-kill curves of NSC319726 were determined for these two isolates. Compared to the untreated conditions, exposure to NSC319726 was associated with significant growth inhibition (Fig. 1). The effect of 4× MIC exposure was greater than that of 2× MIC exposure (*t* test, *P* = 0.04 at 24 h), but the drug remained only fungistatic, since the $\log_{10}$ CFU per milliliter decrease was <99.9%, compared to the starting inoculum.

In checkerboard testing, the interactions between NSC319726 and fluconazole, amphotericin B, and micafungin were defined as indifferent, with fractional inhibitory concentration (FICI) values of 2, 2, and 0.625 and 2, 2, and 2 for isolates IV.1 and I.3, respectively.

The safety and efficacy of NSC319726 were tested in a *Galleria mellonella* model of *C. auris* invasive candidiasis. Escalating single doses of NSC319726 up to 24 mg/kg were well tolerated by the uninfected larvae (100% survival until 7 days following injection). Doses of 6 and 12 mg/kg were selected for the *G. mellonella* model of *C. auris* infection. NSC319726 was effective in rescuing larvae infected with *C. auris* I.3 and IV.1 strains, with significantly improved survival rates, compared to the untreated group (Fig. 2). The doses of 6 and 12 mg/kg were equally effective.

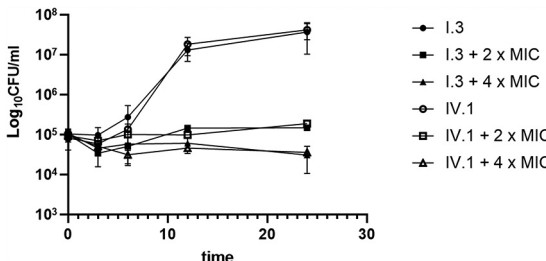

**FIG 1** Time-kill curves for NSC319726 against *Candida auris* isolates I.3 and IV.1. Plots of the $\log_{10}$ CFU/ml of *C. auris* over the time of drug exposure are shown. The concentration of NSC319726 was equal to 2× or 4× the MIC value of each isolate. Results are the mean of biological duplicates with standard deviations (error bars).

## DISCUSSION

NSC319726 is a thiosemicarbazone with zinc-chelating and redox properties that has been investigated for its potential in anticancer therapy and also has demonstrated antibacterial and antifungal activity (9, 10, 12). In *Candida albicans*, NSC319726 was shown to cripple fungal ribosome biogenesis and protein synthesis (10). Reactive oxygen species (ROS) production may also contribute to its antifungal effect (10).

In the present study, we showed that NSC319726 was active against a collection of *C. auris* isolates from different clades, including isolates with presumed resistance to

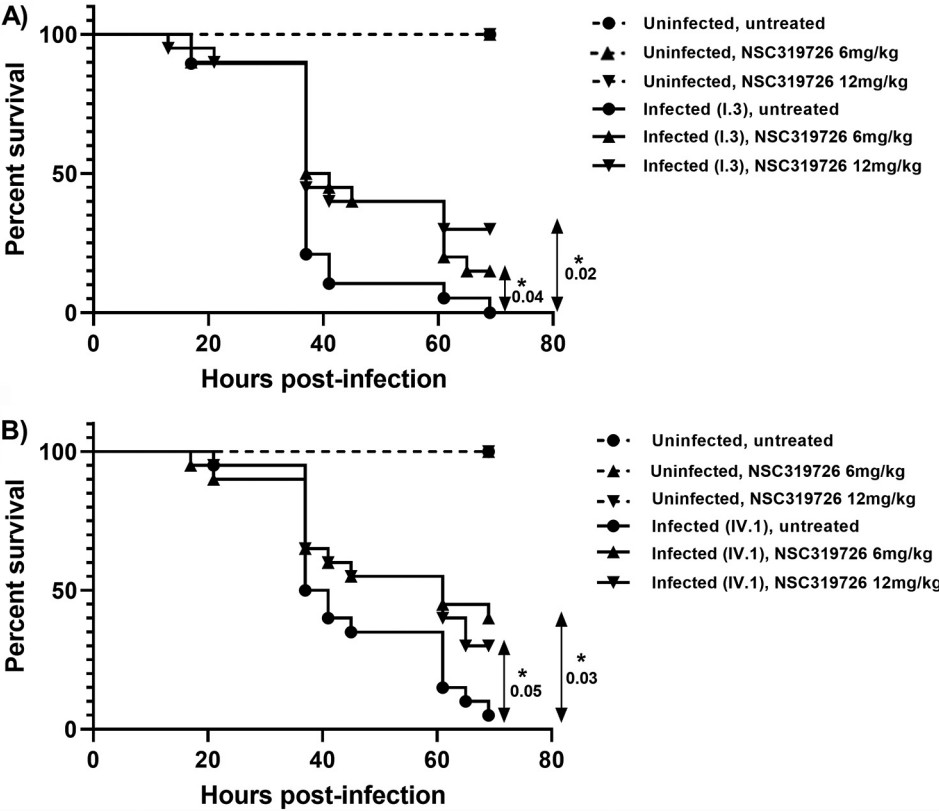

**FIG 2** Effects of NSC319726 against *C. auris* strains I.3 (A) and IV.1 (B) in a *Galleria mellonella* model of invasive candidiasis. Larvae were infected with $5 \times 10^5$ *C. auris* cells in a volume of 40 μl PBS (except for the uninfected control arms, which were treated with 40 μl PBS alone). Two hours later, larvae were injected with 40 μl of NSC319726 at a concentration of 6 mg/kg or 12 mg/kg or in the absence of drug (PBS alone with an equivalent DMSO concentration, i.e., 0.9%). All arms consisted of 20 larvae, except for the uninfected control arms (n = 10). *P* values (log-rank test) are expressed for the comparisons between the infected untreated arm and the infected arm treated with 6 mg/kg of NSC319726 and the infected arm treated with 12 mg/kg of NSC319726. *, *P* values of ≤0.05 were considered significant.

the current antifungal classes. The MIC range of NSC319726 against *C. auris* (0.125 to 0.25 $\mu$g/ml) was similar to those previously reported for other *Candida* spp. (10). Although 100% growth inhibition was observed in broth microdilution assays, time-kill curves showed that the antifungal effect of NSC319726 was fungistatic at concentrations up to 4 times the MIC value. In contrast, Sun et al. reported a fungicidal effect against *C. albicans* (10). They also described synergistic interactions of NSC319726 with fluconazole and caspofungin against *C. albicans* (10), while we observed indifferent interactions with all three antifungal drug classes against two *C. auris* isolates. Finally, we showed that NSC319726 was well tolerated and effective at 6 and 12 mg/kg to treat invasive *C. auris* infection in an invertebrate model of *G. mellonella*. It is noteworthy that, despite a significant impact on survival rates, compared to the untreated group, the rates of survival of the larvae at 72 h after a single injection of NSC319726 were relatively low (30 to 40%). This excessive mortality rate could be due to the persistence of residual fungal growth and infection because of the merely fungistatic effect of the drug. Other animal models (e.g., infections in mice) should be used to assess whether better survival rates could be achieved with repeated dosing. NSC319726 showed minimal toxicity in mice at a dosage of 5 mg/kg/day and *in vitro* in human liver cell lines (9, 10), which is encouraging for further investigations regarding the potential clinical perspectives on this drug.

## MATERIALS AND METHODS

NSC319726 and micafungin (Selleck Chemicals, Houston, TX) and fluconazole, voriconazole, and amphotericin B (Sigma-Aldrich, St. Louis, MO) were obtained as powders and diluted in dimethyl sulfoxide (DMSO) for stock solutions of 10 mg/ml for fluconazole and NSC319726 and 1 mg/ml for the other drugs.

Antifungal susceptibility testing was performed in duplicate by the broth microdilution method according to the Clinical and Laboratory Standards Institute (CLSI) procedure (11). Testing was performed in 96-well U-shaped-well plates, with each well containing a final concentration of $2 \times 10^3$ CFU per ml in 200 $\mu$l of RPMI 1640 medium with 0.2% glucose and glutamine and without bicarbonate, buffered with 0.165 M 3-(*N*-morpholino)propanesulfonic acid (MOPS) to pH 7, including a control well (no drug) and wells containing the drug at the desired concentration (the ranges of concentrations with doubling dilutions were 0.125 to 64 $\mu$g/ml for fluconazole, 0.03 to 16 $\mu$g/ml for voriconazole and amphotericin B, 0.015 to 8 $\mu$g/ml for micafungin, and 0.015 to 16 $\mu$g/ml for NSC319726). Plates were incubated at 35°C for 24 h before reading. MICs were defined as the concentrations achieving 100% growth inhibition (i.e., no residual growth, by visual inspection) for amphotericin B and NSC319726 and $\geq$50% inhibition for micafungin, fluconazole, and voriconazole. For each drug, the MIC$_{50}$ and MIC$_{90}$ were defined as the concentrations at which 50% and 90% of the isolates, respectively, were inhibited.

Time-kill curves were determined as described previously (13). Strains were grown overnight in liquid yeast extract-peptone-dextrose (YEPD). The suspension was diluted to a concentration of $10^5$ CFU/ml in 10 ml RPMI 1640 medium and incubated under constant agitation (220 rpm) at 37°C in the absence of drug and in the presence of NSC319726 at concentrations of 2× and 4× MIC. For each time point (0, 3, 6, 12, and 24 h), aliquots of 100 $\mu$l were diluted in phosphate-buffered saline (PBS) and 10 $\mu$l was spread on YEPD agar plates. CFU were counted after 24 to 48 h of incubation at 37°C. The experiment was performed in biological and technical duplicates.

Drug interactions between NSC319726 and fluconazole, amphotericin B, and micafungin were tested by checkerboard dilution to determine the FICI, as described previously (14). Synergism, indifference, and antagonism were defined as FICI values of $\leq$0.5, >0.5 to 4, and >4, respectively.

For the invertebrate model, *Galleria mellonella* larvae (BioSystems Technology Ltd., University of Exeter, Exeter, UK) weighing 300 to 400 mg were used. The tolerability of escalating doses of NSC319726 (3, 6, 12, and 24 mg/kg) was assessed in groups of 5 larvae in order to determine the optimal dosage. For the invasive candidiasis model, groups of 20 larvae were injected with 40 $\mu$l PBS solution containing $5 \times 10^5$ cells of the *C. auris* strain I.3 or IV.1 or with PBS alone (control groups of 10 larvae). Two hours later, larvae were injected with 40 $\mu$l PBS containing 6 mg/kg or 12 mg/kg NSC319726 or without the drug (i.e., with an equivalent concentration [0.9%] of DMSO). Larvae were kept in the dark at 37°C and monitored for survival three times daily up to 72 h. Data were analyzed by the log-rank test (GraphPad Prism software).

## ACKNOWLEDGMENTS

We thank Marie-Elisabeth Bougnoux from Institut Pasteur (Paris, France), Maurizio Sanguinetti from Università Cattolica del Sacro Cuore (Rome, Italy), Judith Berman and Ronen Ben-Ami from Tel Aviv University (Tel Aviv, Israel), Guillermo Garcia-Effron from Universidad Nacional del Litoral (Santa Fe de la Vera Cruz, Argentina), Nicolas Papon from Angers University (Angers, France), and Arnaud Riat from Geneva University Hospital (Geneva, Switzerland) for providing us with all of the *Candida auris* strains in this study.

This work was supported by a research grant from the Santos-Suarez Foundation.

F.L. received research grants from The Swiss National Science Foundation, Novartis, MSD, and Pfizer outside the submitted work and speaker honoraria from Gilead.

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
