## [Reviewer comments · Microbiology Spectrum]

Microbiology Spectrum

Assessment of the *in vitro* and *in vivo* antifungal activity of NSC319726 against *Candida auris*

Jizhou Li, Alix Coste, Daniel Bachmann, Dominique Sanglard, and Frederic Lamoth

Corresponding Author(s): Frederic Lamoth, Lausanne University Hospital

Review Timeline:

Submission Date:	August 25, 2021
Editorial Decision:	September 10, 2021
Revision Received:	September 23, 2021
Accepted:	September 23, 2021

Editor: Teresa O'Meara

Reviewer(s): The reviewers have opted to remain anonymous.

Transaction Report:

DOI: <https://doi.org/10.1128/Spectrum.01395-21>

September 10, 2021

Dr. Frederic Lamoth
Lausanne University Hospital
Lausanne
Switzerland

Re: Spectrum01395-21 (Assessment of the in vitro and in vivo antifungal activity of NSC319726 against *Candida auris*)

Dear Dr. Frederic Lamoth:

Thank you for submitting your manuscript to Microbiology Spectrum.

As you can see below, both reviewers were generally in favor of the manuscript and would like to see only minor modifications before it can be published. Both reviewers were in favor of moving the supplemental figure to the main text, and both would like to see more methodological details on some of the experiments.

When submitting the revised version of your paper, please provide (1) point-by-point responses to the issues raised by the reviewers as file type "Response to Reviewers," not in your cover letter, and (2) a PDF file that indicates the changes from the original submission (by highlighting or underlining the changes) as file type "Marked Up Manuscript - For Review Only". Please use this link to submit your revised manuscript - we strongly recommend that you submit your paper within the next 60 days or reach out to me. Detailed information on submitting your revised paper are below.

Link Not Available

Sincerely,

Teresa O'Meara

Journals Department
Reviewer comments:

Reviewer #1 (Comments for the Author):

This manuscript describes the antifungal activity of an investigational chemotherapeutic NSC319726 against *Candida auris* clinical isolates both in vitro and in vivo. Overall, while the antifungal activity of this agent has been described previously, as mentioned by the authors of this manuscript, these studies expand what is known to an emerging healthcare associated and multidrug resistant pathogen of great clinical concern. The manuscript is well written and the procedures are easy to follow and understand.

-In the methods the authors describe that MIC testing was performed per CLSI methods, however the description of the endpoint used for micafungin was complete growth inhibition rather than the ~50% inhibition as detailed by CLSI M27 S4. If the authors did indeed use an alternative and more stringent endpoint for micafungin than is standard, this should be clearly mentioned and justification for the decision to do so detailed.

-The authors should include a description of the media, incubation time, and incubation temperatures used for MIC testing particularly if they differ from the CLSI M27 S4 as with the previous point. These parameters all can significantly impact MIC determination, particularly with *Candida auris*, and as the comparative MIC values are a key aspect of the manuscript they should be as clearly defined/ described as possible.

-The supplementary figure should be moved to the main text. The finding that this investigational agent is fungistatic against *Candida auris* but was previously shown to be fungicidal against *Candida albicans* is important and of interest.

-The arrows denoting the significance of comparisons between treatments in Figure 1 are a bit confusing as they are placed between the endpoint of the graph and the legend. These should be moved or changed to more clearly show which of the two they are corresponding to.

-In line 35-36, the authors describe the clades of *C. auris* to be geographic clades. While references like this are common, this is somewhat misleading. While the clades were each associated with a geographic region where they were first found, assignment of isolates to clades is done based up on genetic relatedness and the global dissemination of *C. auris* has resulted in isolates from various clades being identified in many countries (as seen in Table 1). It would seem simply changing the word "geographic" to "genetic" would be more appropriate.

-lines 82 through 86 are very redundant and should be revised/ reworded to address this.

Reviewer #2 (Comments for the Author):

The manuscript is well written and presents novel preliminary data on NSC319726 activity against *Candida auris*.

Please see my detailed comments below.

1. It seems the tested drug concentration ranges were rather unusual - judging by fluconazole and micafungin MICs {greater than or equal to}1024. Please clarify (in the Materials and Methods section) what drug concentration ranges were applied for each drug.
2. The time-kill curves should be performed in biological triplicates, and error bars presented on the graph. Moreover, it is not clear why the graph presenting time-kill curves was made a Supplementary Material and not a Figure in the main text.
3. Please provide some explanation why stains I.3 and IV.1 were selected for in vivo studies.
4. Please provide a comment (in the discussion) on the in vivo results in light of the discovery of fungistatic activity of NSC319726.
5. Lines 36-37: grammar; should be: South Asian, South African, South American.

Staff Comments:

Preparing Revision Guidelines

For complete guidelines on revision requirements, please see the journal Submission and Review Process requirements at <https://journals.asm.org/journal/Spectrum/submission-review-process>.

Submissions of a paper that does not conform to Microbiology Spectrum guidelines will delay acceptance of your manuscript. "

Please return the manuscript within 60 days; if you cannot complete the modification within this time period, please contact me. If you do not wish to modify the manuscript and prefer to submit it to another journal, please notify me of your decision immediately so that the manuscript may be formally withdrawn from consideration by Microbiology Spectrum.

If your manuscript is accepted for publication, you will be contacted separately about payment when the proofs are issued; please follow the instructions in that e-mail. Arrangements for payment

must be made before your article is published. For a complete list of **Publication Fees**, including supplemental material costs, please visit our website.

Spectrum 01395-21 : responses to reviewers comments

Line numbering refers to the "clean" revised manuscript.

Reviewer #1 (Comments for the Author):

-In the methods the authors describe that MIC testing was performed per CLSI methods, however the description of the endpoint used for micafungin was complete growth inhibition rather than the ~50% inhibition as detailed by CLSI M27 S4. If the authors did indeed use an alternative and more stringent endpoint for micafungin than is standard, this should be clearly mentioned and justification for the decision to do so detailed.

Response: We agree that the CLSI M27 S4 document defines the cut-off at 50% inhibition for micafungin. Actually, we did not observe any trailing effect regarding micafungin (clear cut inhibition) for all isolates, which means that the MIC 50% inhibition and MIC 100% inhibition are the same. We agree that strict adherence to CLSI method is warranted and we have modified the MIC definitions for micafungin (50% instead of complete) in methods (lines 126-129), which actually does not affect the results presented in text and in Table 1.

-The authors should include a description of the media, incubation time, and incubation temperatures used for MIC testing particularly if they differ from the CLSI M27 S4 as with the previous point. These parameters all can significantly impact MIC determination, particularly with *Candida auris*, and as the comparative MIC values are a key aspect of the manuscript they should be as clearly defined/ described as possible.

Response: We have adhered to CLSI procedure. We have now added details about media, inocula, incubation time and temperature in methods (lines 118-129).

-The supplementary figure should be moved to the main text. The finding that this investigational agent is fungistatic against *Candida auris* but was previously shown to be fungicidal against *Candida albicans* is important and of interest.

Response: We agree. The paper was initially submitted as an "observation", which according to journal guidelines allow only 2 figures/tables. This is why we have moved the time-kill curve as Supplemental material. We agree that it makes perfect sense to move it as a figure in the main manuscript (now inserted as new Figure 1).

-The arrows denoting the significance of comparisons between treatments in Figure 1 are a bit confusing as they are placed between the endpoint of the graph and the legend. These should be moved or changed to more clearly show which of the two they are corresponding to.

Response: We have moved the arrows in the graph to clearly show the curves that have been compared. We have also added more precisions in the legend.

-In line 35-36, the authors describe the clades of *C. auris* to be geographic clades. While references like this are common, this is somewhat misleading. While the clades were each associated with a geographic region where they were first found, assignment of isolates to clades is done based on genetic relatedness and the global dissemination of *C. auris* has resulted in isolates from various clades being identified in many countries (as seen in Table 1). It would seem simply changing the word "geographic" to "genetic" would be more appropriate.

Response: We agree and we have changed the sentence as follows: five genetic clades have been identified from distinct geographical origin: South Asian (1), etc... (lines 36-38).

-lines 82 through 86 are very redundant and should be revised/ reworded to address this.

Response: We agree. We have suppressed the first sentence.

Reviewer #2 (Comments for the Author):

1. It seems the tested drug concentration ranges were rather unusual - judging by fluconazole and micafungin MICs {greater than or equal to}1024. Please clarify (in the Materials and Methods section) what drug concentration ranges were applied for each drug.

Response: We have indeed tested some higher concentrations than those recommended in the range of the CLSI procedure. For a strict adherence to CLSI recommendations (as also suggested by Reviewer 1), we have now provided only the results within the CLSI ranges (see modifications in methods, lines 123-125, results lines 53-56, and Table 1).

2. The time-kill curves should be performed in biological triplicates, and error bars presented on the graph. Moreover, it is not clear why the graph presenting time-kill curves was made a Supplementary Material and not a Figure in the main text.

Response: We agree. The initial figure was the results of technical duplicates but a single biological replicate. We have now repeated the experiment in biological duplicates and technical duplicates for each of them (see methods lines 137-138) and results have been provided as mean of the biological duplicates with error bars (see new Figure 1 and legend). Note that, because this is a logarithmic scale, error bars are not visible for some points when it is a very narrow range. Because the results of the two biological duplicates of each strain, as well as the comparison between the two different strains, were reproducible, we did not perform a third biological replicate. The figure was initially put in the supplemental material because we have submitted this manuscript as an "observation", for which only two tables/figures are allowed according to the policy of the journal. However, we fully agree that it makes more sense to shift this figure to be published within the manuscript (now inserted as new Figure 1).

3. Please provide some explanation why strains I.3 and IV.1 were selected for in vivo studies.

Response: strains I.3 and IV.1 were selected because they represent the two most important clades (I, South Asian and IV, South American) and different profiles of susceptibility to standard antifungals: one isolate susceptible to fluconazole and echinocandins (IV.1) and one resistant to both drugs (I.3). We have added some explanation (lines 62-65).

4. Please provide a comment (in the discussion) on the in vivo results in light of the discovery of fungistatic activity of NSC319726.

Response: NSC319726 was fungistatic, but the in vitro effect was potent with an important decrease in CFU/ml in time kill curves and a complete visual growth inhibition in AST. The effect in the Galleria model was significant, but possibly not satisfactory in terms of survival (only 30-40% survival at 72h). This could be due to the persistence of residual fungal growth and infection following a single drug injection in the Galleria model because of the only fungistatic effect of the drug. Therefore, we think that further experiments in murine models (with repeated dosing) would be warranted to assess the

actual potential of this drug for a possible future clinical application. We have added some sentences about this in the discussion (lines 100-106).

5. Lines 36-37: grammar; should be: South Asian, South African, South American.

Response: This has been corrected.

September 23, 2021

Dr. Frederic Lamoth
Lausanne University Hospital
Lausanne
Switzerland

Re: Spectrum01395-21R1 (Assessment of the in vitro and in vivo antifungal activity of NSC319726 against *Candida auris*)

Dear Dr. Frederic Lamoth:

Thank you for resubmitting your revised manuscript. All of the reviewer's previous comments were addressed.

Your manuscript has been accepted, and I am forwarding it to the ASM Journals Department for publication. You will be notified when your proofs are ready to be viewed.

Sincerely,

Teresa O'Meara
Editor, Microbiology Spectrum
